# Hyperconnected Openings Codified in a Max Tree Structure: An Application for Skull-Stripping in Brain MRI T1

**DOI:** 10.3390/s22041378

**Published:** 2022-02-11

**Authors:** Carlos Paredes-Orta, Jorge Domingo Mendiola-Santibañez, Danjela Ibrahimi, Juvenal Rodríguez-Reséndiz, Germán Díaz-Florez, Carlos Alberto Olvera-Olvera

**Affiliations:** 1Conacyt-Centro de Investigaciones en Óptica, Aguascalientes 20200, Mexico; cparedes@cio.mx; 2Facultad de Ingeniería, Universidad Autónoma de Querétaro, Santiago de Querétaro 76010, Mexico; danjela.ibrahimi@uaq.mx (D.I.); juvenal@uaq.edu.mx (J.R.-R.); 3Facultad de Medicina, Universidad Autónoma de Querétaro, Santiago de Querétaro 76176, Mexico; 4Unidad Académica de Ingeniería Eléctrica, Universidad Autónoma de Zacatecas, Zacatecas 98000, Mexico; germandiazflorez@gmail.com (G.D.-F.); colvera@uaz.edu.mx (C.A.O.-O.)

**Keywords:** brain segmentation, computer vision, biomedical image processing and understanding, connected openings, hyperconnectivity, regional maxima, viscous transformations

## Abstract

This article presents two procedures involving a maximal hyperconnected function and a hyperconnected lower leveling to segment the brain in a magnetic resonance imaging T1 weighted using new openings on a max-tree structure. The openings are hyperconnected and are viscous transformations. The first procedure considers finding the higher hyperconnected maximum by using an increasing criterion that plays a central role during segmentation. The second procedure utilizes hyperconnected lower leveling, which acts as a marker, controlling the reconstruction process into the mask. As a result, the proposal allows an efficient segmentation of the brain to be obtained. In total, 38 magnetic resonance T1-weighted images obtained from the Internet Brain Segmentation Repository are segmented. The Jaccard and Dice indices are computed, compared, and validated with the efficiency of the Brain Extraction Tool software and other algorithms provided in the literature.

## 1. Introduction

Magnetic resonance imaging (MRI) allows the noninvasive assessment of the patient, and it is useful for early diagnosis, medical monitoring, and the detection of many diseases, such as Alzheimer’s disease, brain aneurysm, brain tumor, and melanoma of the eye [1]. However, there are many applications related to brain imaging that require accurate brain segmentation to separate the skull, scalp, dura, eyes, etc. This procedure is known as skull stripping [2,3]. Several applications of brain segmentation include brain volume estimation [4], image registration [5], automatic tumor detection [6], the first state in cortical flattening procedures [7], and structural studies [8,9,10].

Algorithms for skull stripping can be placed into six broad categories mentioned in the literature [1,11]. The categories include: mathematical morphology, pixel intensity analysis, deformable surfaces, brain atlas, hybrid approaches, and deep learning focus.

Furthermore, there are several software programs designed to complete the skull-stripping task. One of the most widely used software programs is BET (Brain Extraction Tool) [12], which quickly processes complete volumes of images. BET evolves a tessellated mesh of triangles to fit the brain’s surface. However, the resulting crude “skull” image contains a relatively large number of false negatives and positives [13]. Considering the disadvantage mentioned above, BET2 is presented as a new software version, an automated tool for extracting mesh surfaces of the brain, the inner and outer skull, and the scalp from an MRI. Ideally, it requires both T1- and T2-weighted anatomical MRIs, each of a <2 mm resolution in each direction. Another BET improvement is presented in [14], where the authors proposed a faster convergence of the algorithm since they enhanced the vertex displacement, added a new search path, and embedded an independent surface reconstruction process. Other popular algorithms used to separate the skull are the following [15]: BSE (Brain Surface Extractor), SPM2 (Statistical Parametric Mapping v2), McStrip (Minneapolis Consensus Strip), ROBEX (robust brain extraction), and TMBE (Threshold Morphologic Brain Extraction). In [16], the authors used the Richardson–Lucy deconvolution, obtaining high-quality results. However, CoLoRS (coherent local intensity rough segmentation) is a new algorithm to segment an MRI that considers intensity inhomogeneity or bias fields presented in MR volumes [17]. The algorithm is based on clustering and a rough sets theory for simultaneous segmentation and bias field correction of brain MR volumes. The clustering technique allows separating or segmenting components in an image, where statistics are used to group and classify considering texture, color, and form factor, among others. In [18], researchers obtained brain tissue using graph theory, supervoxels, and filtering. In [19], the authors explain the evolution of computational methods in human brain connectivity from 1990 to the present, focusing on graph theory. Graph theory has become a powerful approach for brain imaging analysis, mainly because of its potential to study dynamic behavior over time and disease-related brain changes [20]. In using graph theory, the first step in creating brain graphs is to define the nodes and edges connecting them. It is worth mentioning that connectivity is a powerful concept because the processed images under this notion preserve the contours, avoiding the creation of new maxima or minima.

This paper provides two new openings and introduces two procedures for separating the brain in the MRI T1. Regardless of the method used to separate the skull and nonbrain tissue in an MRI, it is necessary to compare the resultant segmentation to evaluate the performance of the proposed method. The Jaccard and Dice [21,22] indices will be used to compare the resulting segmentations.

The proposals use hyperconnectivity [23], the max-tree [24], viscous transformations [25], and lower levelings [26]. To introduce morphological transformations, Section 2 provides some notions of morphological filtering, connectivity, and hyperconnectivity. Such concepts are explained in detail, and several images illustrate how they operate practically. Section 3 gives the mathematical formalism of the openings defined under the hyperconnectivity. The proposals given in this section are also explained widely to follow the ideas related to mathematical morphology. Section 3.1 shows the criterion based on the maxima of the image to detect the maximum hyperconnectivity, and Section 3.2 formalizes the use of lower leveling applied from the higher extreme. Section 4 illustrates the application of hyperconnected viscous transformations, which allow the skull stripping of the 38 MRI T1 images provided by the IBSR [27]. Once the brain is segmented, the indices of Jaccard and Dice [21,22] are computed and compared with the information obtained from the current literature. Section 5 corresponds to the conclusions.

## 2. Some Basic Concepts of Morphological Filtering and Connections

### 2.1. Basic Notions of Morphological Filtering

Figure 1 shows the elementary structuring element in 3D containing its origin used in this study. $\overset{˘}{B}=\{-x:x\in B\}$ denotes the transposed set with respect to its origin, and λ is a homothetic parameter. The morphological opening, closing, erosion, and dilation transformations are expressed as, $\gamma_{\lambda B}$, $\varphi_{\lambda B}$, $\varepsilon_{\lambda B}$, and $\delta_{\lambda B}$, respectively [28]. Furthermore, the opening ${\tilde{\gamma }}_{\mu }\left(f\right)=R_{f}\left(\varepsilon \right)\left(x\right)$ and closing ${\tilde{\varphi }}_{\mu }\left(f\right)=R_{f}\left(\delta \right)\left(x\right)$ by reconstruction propagate a marker to filter components by size without introducing new contours [29], where $R_{f}$ represents the reconstruction transformation.

### 2.2. Connectivity

One of the most interesting concepts proposed in mathematical morphology is the connected class introduced by Serra [30]. The transformations defined under specific connectivity do not modify the shapes during the processing, and fused regions preserve the contours of the original image. The connected opening $\gamma_{x}\left(A\right)$ is used in practice to separate each one of the components. Figure 2 clarifies this concept.

### 2.3. Viscous Opening

Serra proposed a connection class on the space generated by dilated [31]. The viscous opening belongs to this class and discovers the connected components eroding the image. This transformation is expressed as follows:(1)$$\gamma_{\delta (x)}=\delta_{\lambda }\gamma_{x}\varepsilon_{\lambda }$$

The number of connected components depends on the viscosity parameter λ. Figure 3a displays the original image composed of three arcwise-connected components or three components at viscosity λ=0. Figure 3b–e show the eroded images using disks of sizes 20, 22, 27, and 36. Then, at viscosity λ = 20, there are 4 connected components, whereas, at viscosity λ = 22, the image has 5 connected components, as exemplified in Figure 3b,c. The image in Figure 3f presents the connected components for viscosity λ=22. However, by considering disks as the elementary shapes of the image, it is not possible to detect six connected components for any viscosity. The solution to this problem is to select the connected components at different viscosities (scales), and one option is the traditional algorithm known as ultimate erosion. This consists of choosing the connected components at a specific viscosity λ, such that the viscosity λ+1 will remove them. Another method to select the connected components is to compute the distance function, as shown in Figure 3g, and detect their maxima. The image in Figure 3h contains the ultimate eroded components for viscosities (sizes)25,34,45,64,66, and 68. Figure 3i illustrates the connected components in the viscous lattice sense. Viscous connectivity is interesting because it exploits the goal in binary image segmentation, which consists of splitting the connected components into a set of elementary shapes.

### 2.4. Morphologically Connected Filtering in Viscous Lattices

Equation (Equation 1) utilizes the marker *x* to detect the image components using the punctual opening $\gamma_{x}\left(X\right)$. However, the trivial opening $\gamma_{O}\left(A\right)$ uses other criteria, for example, area, the size of the structuring element, or volume. This opening is presented as follows: (2)$$\gamma_{O}\left(A\right)=\{\begin{array}{cc} A & \mathrm{if}\;A\;\mathrm{satisfies}\;\mathrm{an}\;\mathrm{increasing}\;\mathrm{criterion}\\ \emptyset & \mathrm{Otherwise} \end{array}$$

The operator $\gamma_{O}\left(A\right)$ detects and recovers all image components, fulfilling an increasing criterion. Then, from Equation (Equation 1), the connected viscous opening is expressed as follows:(3)$${\tilde{\gamma }}_{\lambda ,O}\left(X\right)=\delta_{\lambda }{\tilde{\gamma }}_{O}\varepsilon_{\lambda }\left(X\right)$$

For the case of functions, it is denoted as:(4)$${\tilde{\gamma }}_{\lambda ,O}(f)=\delta_{\lambda }{\tilde{\gamma }}_{O}\varepsilon_{\lambda }\left(f\right)$$

Equations (Equation 3) and (Equation 4) permit different viscous openings depending on the increasing criteria.

### 2.5. Hyperconnectivity

Serra introduced the hyperconnectivity concept [23], which permits working with joined or overlapped components. Figure 4 helps to understand this notion for the 2D case. Notice that the eyes link the brain and the skull in Figure 4a, i.e., they are hyperconnected because they form a unique component. Figure 4b presents the regional maxima obtained from Figure 4a. Each of these maxima are located on the brain, eyes, white matter, or skull.

From here, we let Max(f) be the set of the maxima of *f*, whereas ${Max}_{k}\left(f\right)$ denotes the set of maxima of *f* at the *k* level.

Serra [23] considered a class of functions f∈F admitting a unique maximal connected component $\left|Max(f)\right|=1$, where the horizontal cross-sections of functions f∈F are connected in such away that functions that admit a unique maximum generate a hyperconnection. An example of this situation is illustrated in Figure 5c,d.

## 3. Proposal of Using Hyperconnections and Viscous Transformations

### 3.1. Hyperconnected Opening

Similarly to Serra, we consider the class *W* of those functions admitting a maximal connected component. Particularly, we define the class *W* as those extracted from a function *f*, containing only one maximum using the reconstruction transformation $R_{f}\left(g\right)\left(x\right)$. We let Max(f) be the set of the maxima of the function *f*. The marker $h_{M_{i}}$ is expressed as follows, $h_{M_{i}}\left(x\right)=f\left(x\right)$ $\forall x\in M_{i}\in Max\left(f\right)$, and otherwise, $h_{M_{i}}\left(x\right)=0$. Thus, a particular set of functions recovered from *f* with a unique maximum is composed of $g_{M_{i}}=R\left(f,h_{i}\right)$ functions. Then, the set *W* is defined as follows: $W_{f}=\left\{g_{M_{i}}:\forall M_{i}\in Max\left(f\right)\right\}$.

Figure 5 exemplifies this situation. Figure 5a shows two overlapping functions, $g_{1}$ and $g_{2}$. For the MRI case, $g_{1}$ represents a maximum on the brain, and $g_{2}$ is a maximum on the skull; however, the regions under the intersection of both functions indicate that the brain and the skull overlap, i.e., they are connected or hyperconnected. Figure 5b illustrates the supremum between $g_{1}$ and $g_{2}$. Note that it is not possible to recover $g_{1}$ or $g_{2}$ from $g_{1}⋁g_{2}$. This is what we visualize in reality; our eyes would observe how the brain and the skull appear in two places in a certain slice; nevertheless, lower slices connect them.

The reconstructed functions correspond to $g_{M_{1}}$ and $g_{M_{2}}$, displayed in Figure 5c,d. These images come from an individual reconstruction using each maximum computed from Figure 5b. Figure 5c,d illustrate how to detect the markers to separate the brain and skull, and Equation (Equation 6) represents it formally. The maximum to be treated is selected and subsequently reconstructed using the transformation by reconstruction *R*. In Figure 5e, the two functions $g_{1}$ and $g_{2}$ do not overlap; hence, both functions can be retrieved. However, this is not what really happens in image segmentation.

Figure 6 illustrates a real example where there are several hyperconnected functions. Figure 6a,b show the original image and its maxima, respectively. In contrast, Figure 6c illustrates the obtention of a hyperconnected function computed from the extreme region marked with a circle of green color. Note that each maximum can be used to produce a hyperconnected function.

Now, we introduce the trivial criterion to build openings as follows:(5)$$\gamma_{O}\left(g_{M_{i}}\right)=\{\begin{array}{c} g_{M_{i}}\;\;\;\;\;\;\;\;\;\;\mathrm{if}\;Vol\left(g_{M_{i}}\right)\geq \mu_{v}\;\\ \;\;\;\;\;\;\;\;\;\;\;\\ 0\;\;\;\;\;\;\;\;\;\;\;\;\;\mathrm{Otherwise}\;\;\;\;\;\;\;\;\; \end{array}$$
where Vol represents the volume, and $\mu_{v}$ denotes the increasing criterion given by volume. The opening $\gamma_{M_{i}}$ is expressed as:(6)$$\gamma_{M_{i}}(f)=g_{M_{i}}=R\left(f,h_{M_{i}}\right)$$

Then, we define Equation (Equation 7) as:(7)$${\tilde{\gamma }}_{\lambda ,O}\left(f\right)=\delta_{\lambda }{\tilde{\gamma }}_{\mu_{v}}\varepsilon_{\lambda }\left(f\right)$$
with
(8)$${\tilde{\gamma }}_{\mu_{v}}\left(f\right)=⋁\{\gamma_{O}\left(g_{M_{i}}\right)\;:\;g_{M_{i}}\in W\left(f\right)\}$$

Equation (Equation 8) specifies that ${\tilde{\gamma }}_{\mu_{v}}$ is obtained from those reconstructed maxima that fulfill the increasing criterion; for this, the supremum operator is necessary.

Formally, the next expression considers the highest maximum, called extreme hyperconnectivity:(9)$$\mu_{v}^{M}=⋁\{Vol(g_{M_{i}})\;:\;M_{i}\in Max\left(f\right)\}$$

In the present work, the following connected viscous opening is used:(10)$${\tilde{\gamma }}_{\mu_{v}^{M}}\left(f\right)={{\delta_{\lambda }\tilde{\gamma }}_{\mu_{v}^{M}}\varepsilon }_{\lambda }\left(f\right)$$

Figure 7 shows an example in 2D using the input image in Figure 7a. The erosion $\varepsilon_{\lambda }\left(f\right)$ is in Figure 7b. The maxima computed using the max-tree can be found in Figure 7c. Figure 7d shows the maximum fulfilling the increasing criterion to compute the most important hyperconnected component, whereas Figure 7e exemplifies the opening ${\tilde{\gamma }}_{\mu_{v}^{M}}$. Figure 7f contains the threshold calculated by the Otsu algorithm. The output image corresponds to the image in Figure 7g, and a mask with the input image permits brain recovery.

### 3.2. Hyperconnected Functions and Lower Leveling

Another useful transformation used in this work to segment the brain is the lower leveling, $\psi_{\mu ,\alpha }^{1}(f,\;g)=f⋀[g⋁(\delta_{\mu }(g)-\alpha )]$ [26]. This transformation works similar to the opening by reconstruction; nonetheless, the α∈[0,255] parameter permits controlling the reconstruction of the marker into the original image.

The proposed marker *g* considers the regional maxima similar to those defined in Section 3.1, and subsequently, it is iterated using the lower leveling transformation. Therefore, following similar steps to deduce Equation (Equation 10), Equation (Equation 11) is obtained:(11)$$\gamma_{\lambda ,O}^{*}=\delta_{\lambda }\gamma_{\mu_{v}}^{*}\varepsilon_{\lambda }\left(f\right)$$
with
$$\gamma_{\mu_{v}}^{*}\left(f\right)=⋁\{\gamma_{O}\left(g_{M_{i}}\right)\;:\;g_{M_{i}}\in W\left(f\right)\}$$

In practice, Equation (Equation 11) indicates that the erosion is computed on the input image to separate connected components. Posteriorly, the marker is established on a specific maximum selected by an increasing criterion. The marker *g* is iterated following the lower leveling operator. During the reconstruction process, the α parameter avoids a complete marker reconstruction into the reference image. In the end, the output image is dilated.

## 4. Results: Brain Extraction Using Hyperconnectivity

### 4.1. Brain Extraction Based on the Maximum Hyperconnected Function

The difficulties found in the skull stripping procedure and its importance have led to the introduction of a wide range of proposals to address the problems with the procedure. A fundamental challenge to segment the brain in an MRI is the connection between the brain and the skull. Based on classical connectivity, the brain and skull make a connected component. Therefore, to overcome this problem, another type of connection must be used: for example, the viscous connectivity to separate them. The MRI dataset of 38 normal subjects processed in this paper comes from the Internet Brain Segmentation Repository (IBSR), developed by the Center for Morphometric Analysis (CMA) at the Massachusetts General Hospital [27]. However, considering that the images of this database contain intense contrast changes between the different three-dimensional image sections, converting our analysis is a real challenge. This problem, linked to the low quality of images, cannot always be avoided. Thus, robust methods must be implemented to process the images to resolve these difficulties. The proposed morphological transformations presented in this work give good results even when working with poor-quality images. The basic idea behind the segmentation problem of MRI images is the use of viscous connected transformations. Instead of computing the classically connected components on the original image, the morphological erosion $\varepsilon_{\lambda }$ provides them (step one in Figure 8), and the following consists of determining the maximum hyperconnected function (step two in the diagram of Figure 8). Morphological dilation $\delta_{\lambda }$ permits the generation of a viscous component in step three, and a threshold based on the Otsu method (step four) enables brain detection. Figure 7 shows the sequence of operators proposed to separate the brain. The Jaccard ($JC=\frac{\left|X⋂Y\right|}{\left|X⋃Y\right|}$) and Dice ($DC=2\frac{\left|X⋂Y\right|}{\left|X|+|Y\right|}$) [21,22] indices are computed to compare our results with those presented in the current literature. The Jaccard and Dice indices were designed to measure the overlap between two given objects and yield a value between zero (no overlap) and one (complete overlap). The metrics are straightforward to compute and interpret, but they should be applied carefully. These indices are not appropriate for use in the following cases [32]: (1) in small segmented structures such as brain lesions, cell images at low magnification, or distant cars; (2) in the presence of noise; (3) they do not have the capability of distinguish differences in shapes; (4) they are not appropriate for detection and localization tasks.

Our interest is evaluating the intersection with the ground truth images provided by databases utilized here. Table 1 displays such indices for the BET algorithm in columns “BET Jaccard” and “BET Dice”, and our results are in columns “MHF Jaccard” and “MHF Dice”. The data reported in Table 1 use 38 volumes of the IBSR repository, and they are plotted in Figure 15.

### 4.2. Brain Extraction Based on Hyperconnected Functions and Lower Leveling

The procedure to separate the brain using hyperconnectivity and the viscous opening $\gamma_{\mu_{v}}^{*}$ is similar to that presented in Section 4.1. The diagram in Figure 9 presents the new opening $\gamma_{\mu_{v}}^{*}$ in step 2. Because the lower leveling is iterated with some α value, the marker propagation stops in the image minima more quickly. In this case, the dura represents a relevant minimum in the picture. Therefore, the skull will never reach high-intensity levels, and as a result, the lower leveling transformation produces an excellent outcome. The last step consists of separating the complete reconstructed brain using the Otsu threshold. Figure 10 presents a set of processed images following the steps given in Figure 9. The Jaccard and Dice indices are computed and presented in Table 2. Thus, Figure 11 shows several brain slices in different planes to appreciate the performance of the hyperconnected opening on a specific volume. Moreover, Figure 12 illustrates how the brain is reconstructed by iterating the hyperconnected leveling transformation with different slopes α.

In addition, Figure 13 shows the performance of the algorithm presented in Figure 8 using a file obtained from the Neurofeedback Skull-stripped (NFBS) repository [33]. Ten brain IMR volumes of the NFBS are processed using the MHF algorithm displayed in Figure 8, and their respective Jaccard and Dice indices can be observed in Table 3. The mean values of Table 3 are displayed in Table 4. Such indices are useful to be compared with other results presented in the current literature.

Table 4 displays the mean values of the Dice and Jaccard indices computed from the number of volumes specified in the last column utilizing different methodologies, and most of them utilize the IBSR dataset. Most articles compare with BET, for which Table 4 includes the indices associated with this methodology.

For the SPM8 Seg, SPM8 VBM, SPM8-NewSeg, FSL, and Brainsuite methods [34], the authors did not report the results for BET. However, by comparing such indices related to the MHF and HLL procedures, a better performance is observed for the algorithms proposed in this paper. The same situation is found when compared to the methodology introduced by Somasundaram et al. [35]. The results from the 10 brain MRIs obtained from the NFBS repository are high because the MRI volumes do not present the eyes, facilitating skull segmentation. Nevertheless, similar performances are presented when MHF and HLL procedures are compared to Zhang et al. [36], Jiang et al. [37], Mendiola et. al. [38], and Galdames et al. [39]. In Jiang et al. [37], the authors use a nonlinear speed function in the hybrid level set model to eliminate boundary leakage. When using the method, an active contour neighborhood model is applied iteratively slice by slice until the neighborhood of the brain boundary is obtained. The results show high values for the computed indices because of the edges enhancement. However, there are two problems: (1) the brain extraction requires a semi-global understanding of the image and (2) the weak boundaries between the brain tissues and surrounding tissues.

Additionally, in Mendiola et al. [38], indices of Jaccard and Dice are high, and so is the algorithm’s execution time. To overcome this disadvantage, in this paper, the max-tree structure, and the hyperconnectivity notion are utilized. The results displayed in Table 5 show two faster procedures with indices comparable to those obtained in [37,38]. The results indicate minimal differences between the methods [37,38], and the paper presented here. The proposal introduced here has the disadvantage that before loading the flat areas to the max-tree, smoothing filtering is performed to reduce the amount of regions that are loaded to the structure, producing the fusion of elongated and narrow regions. Furthermore, separately comparing the indices of the averages of the 18 (IBSR1) and 20 (IBSR2) volumes, our procedure overcame the indices reported in [37] for the IBSR1 data set and are similar to the IBSR2. Furthermore, the execution time of our proposal can be found in Table 5, together with the times reported in [38,40].

Nonetheless, as noted in Table 4, the BET method presents low indices when compared to other methodologies. To better appreciate the data provided in Table 4, Figure 14 presents such information. BET is used widely around the world for the following reasons: (i) Usually, researchers do not need complete brain segmentation, and (ii) it is quicker. The computation time, which is not commonly reported in papers, should be considered as an important parameter. With this in mind, Table 5 indicates that the computation time of the proposed HLL method is close to that of the BET algorithm, but with a high degree of efficiency in brain segmentation. This situation is illustrated clearly in Figure 15, where MHF and HLL algorithms outperform BET. The BET algorithm produces approximated segmentations. However, sometimes, when images contain important variations in illumination, brain segmentation is similar to a sphere [38].

## 5. Conclusions

The two methods proposed in this paper utilize hyperconnectivity and viscous lattices, which permit separating the brain, even with poor-quality images. A criterion of maximum hyperconnectivity to extract the main component (the brain) and a threshold based on the Otsu method guarantee automation in the process. The efficiency related to segmentation is better than that of the BET algorithm and similar to the results shown in [36,37,38,39]. Moreover, the computation time is comparable to the BET and quicker than that of the method presented in [38]. However, segmenting hyperconnected components entails separating overlapping parts. If the pixel intensity level between the elements is notable, the possibility of obtaining the right segmentation is high; otherwise, it is necessary to use other criteria to separate components mainly during the reconstruction process. Moreover, the computed indices for volumes of teh NFBS repository are high because the IMR volumes lack eyes, and the segmentation is simple. Furthermore, the better execution time to separate brain corresponds to our proposal when compared with those reported in [37,38], whereas the Jaccard and Dice indices are similar.

Three things should be considered in future work: (1) hippocampal segmentation, where the intensity levels among the regions are similar; (2) white and gray matter separation, since they are related to neural damage and memory problems due to aging; and (3) the proposed transformations work adequately for T1 images, and not for other modalities. This inconvenience will be solved.

## Figures and Tables

**Figure 1 sensors-22-01378-f001:**
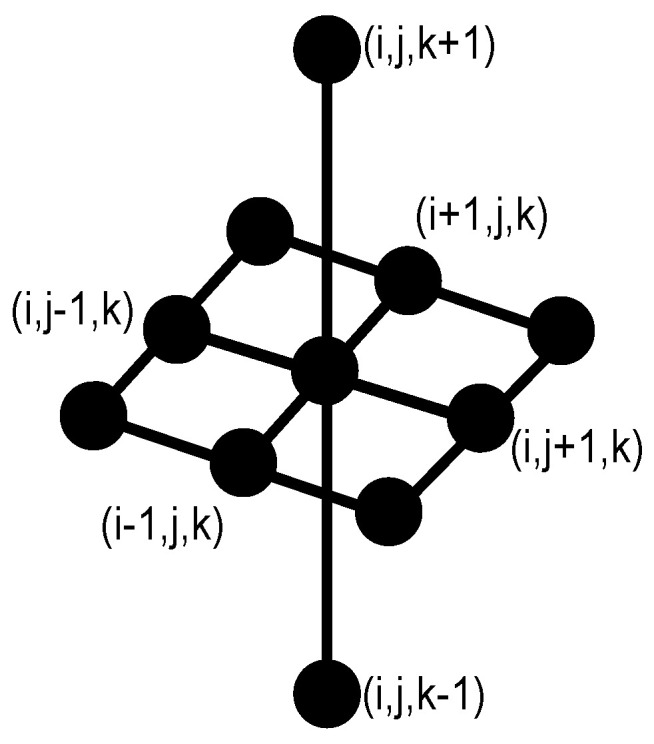
Three-dimensional structuring element *B* with 11 neighbors. This configuration allows processing 11 pixels into 3 slices. The nine points are taken from the central image, and the two points remaining take information from neighbor images.

**Figure 2 sensors-22-01378-f002:**
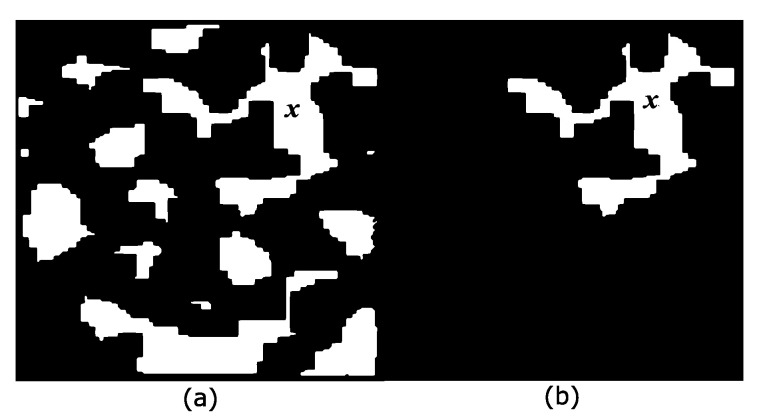
Punctual opening. (**a**) Original image *X* and the marker *x*; (**b**) The punctual opening $\gamma_{x}\left(X\right)$ extracts the component where marker *x* is located.

**Figure 3 sensors-22-01378-f003:**
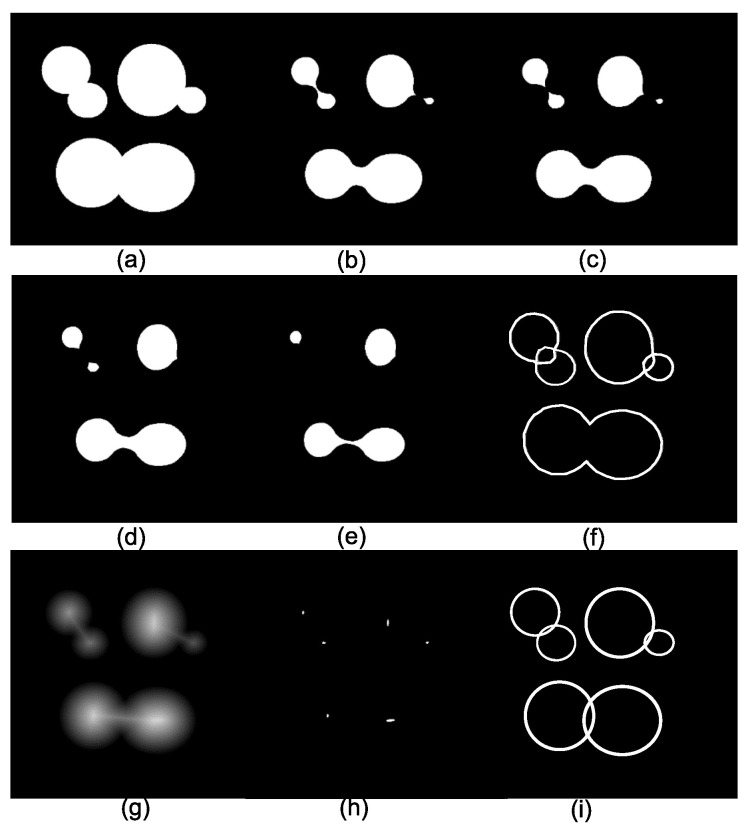
Connected components in viscous lattices. (**a**) The original set; (**b**–**e**) eroded images by disks of sizes 20, 22, 27, and 36; (**f**) connected components for viscosity λ = 22; (**g**) distance function; (**h**) ultimate eroded components for viscosities (sizes) 25, 34, 45, 64, 66, and 68; and (**i**) connected components in the viscous lattice sense.

**Figure 4 sensors-22-01378-f004:**
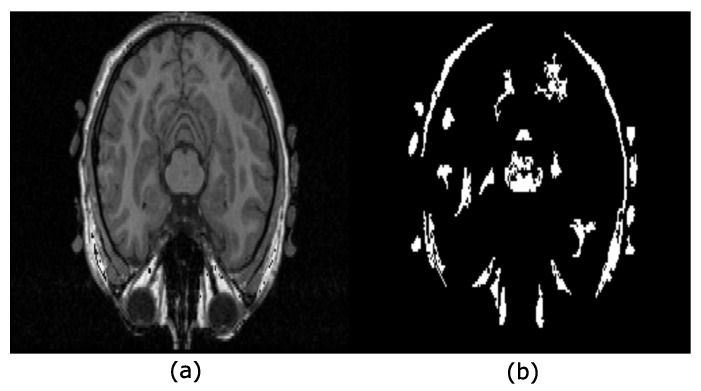
Hyperconnectivity concept. (**a**) The image shows a hyperconnection because all the structures into the head overlap, forming a unique component; (**b**) regional maxima detected from the image in (**a**) after applying the filter ${\tilde{\gamma }}_{\mu =2}{\tilde{\varphi }}_{\mu =2}\left(f\right)$.

**Figure 5 sensors-22-01378-f005:**
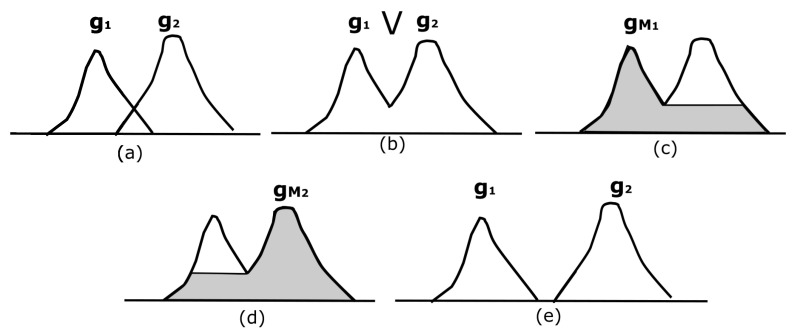
Hyperconnected functions. (**a**) Two functions $g_{1}$ and $g_{2}$; (**b**) supremum of the functions in (**a**); (**c**,**d**) display two hyperconnected functions with a maximum; (**e**) trivial hyperconnected functions $g_{1}$ and $g_{2}$.

**Figure 6 sensors-22-01378-f006:**
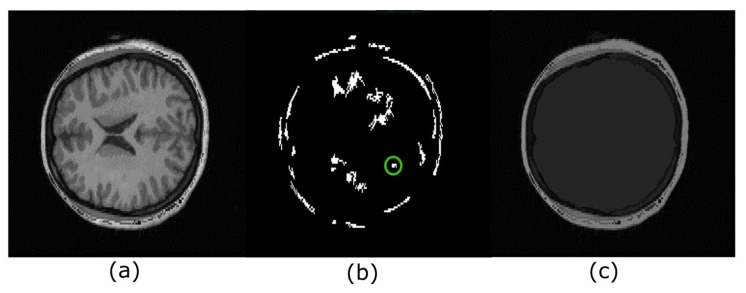
Hyperconnected functions. (**a**) Original image; (**b**) each regional maximum detected corresponds to different regions within the brain, and each maximum can be used to recover other internal structures. However, our work currently investigates this case. The maximum in a green circle is the marker to produce the image in (**c**); (**c**) hyperconnected function associated with regional maximum marked with the green circle.

**Figure 7 sensors-22-01378-f007:**
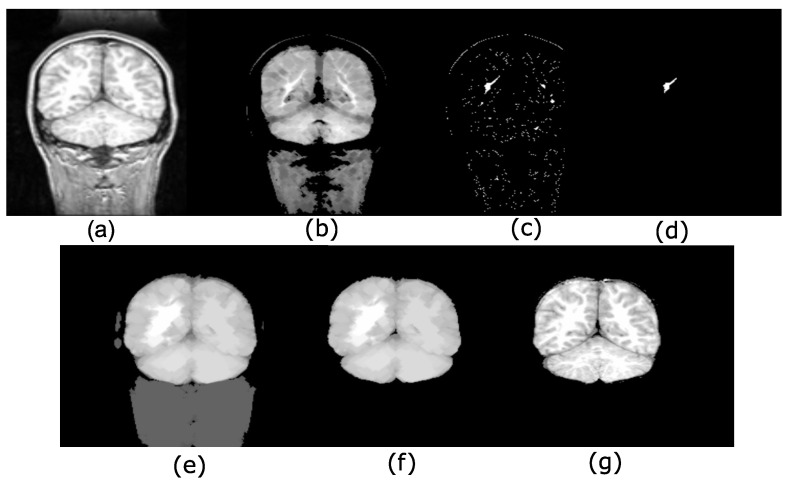
Maximum hyperconnectivity and ultimate connected viscous opening example. (**a**) Original image; (**b**) $\varepsilon_{\lambda }\left(f\right)$ size λ=3. The morphological erosion separates brain and skull; (**c**) maxima detection; (**d**) maximum fulfilling the increasing criterion, i.e., the greater volume obtained of a max-tree branch; (**e**) ${\delta_{\lambda =3}\tilde{\gamma }}_{\mu_{v}^{M}}\varepsilon_{\lambda =3}\left(f\right)$; (**f**) Otsu threshold to eliminate low-intensity levels; (**g**) mask with the original image.

**Figure 8 sensors-22-01378-f008:**
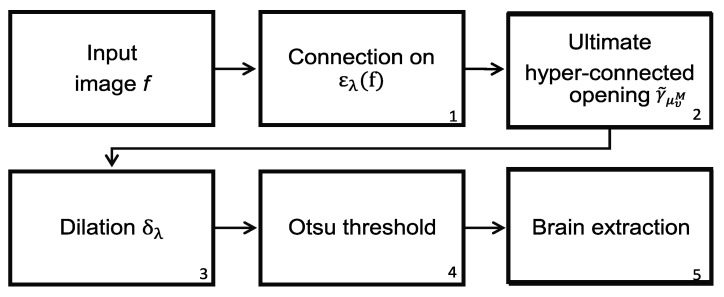
Steps to segment the brain using the maximum hyperconected function (MHF), denoted as the MHF procedure.

**Figure 9 sensors-22-01378-f009:**
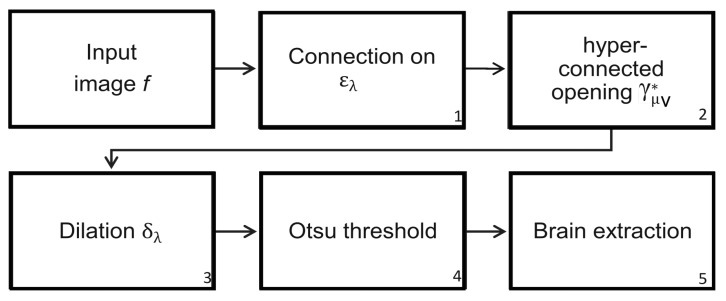
Steps to segment the brain using the Hyperconnected Lower Leveling (HLL) procedure.

**Figure 10 sensors-22-01378-f010:**
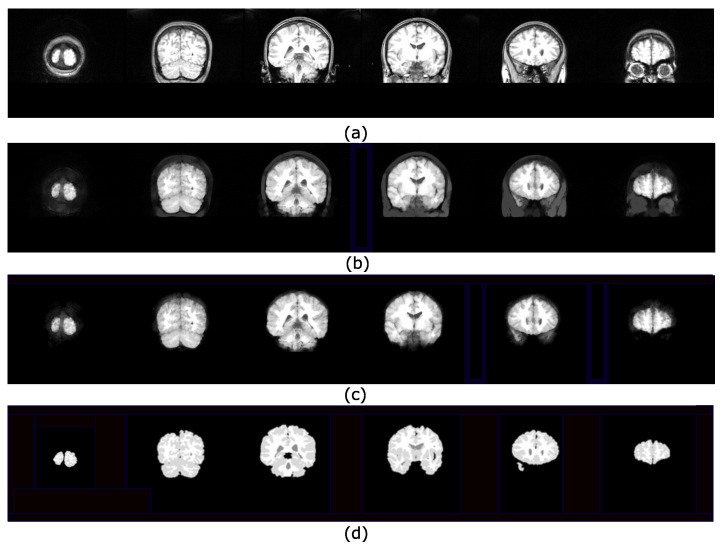
Set of images processed using the HLL procedure presented in Figure 9: (**a**) Input images; (**b**) viscous components obtained from step (3) using α=3,
λ=3 and the greater volume computed from a max-tree branch; (**c**) brain segmentation after applying the Otsu threshold; (**d**) ground truth images obtained from the IBSR repository.

**Figure 11 sensors-22-01378-f011:**
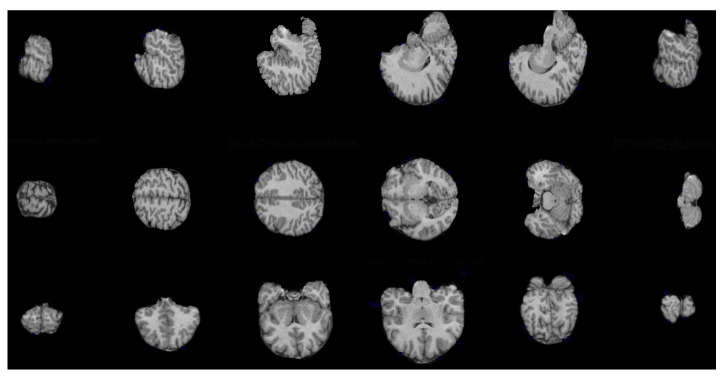
Brain slices in axial, sagittal, and coronal planes obtained by applying the hyperconnected viscous opening $\gamma_{\mu_{v}}^{*}$, with $\varepsilon_{\lambda =3}\left(f\right)$.

**Figure 12 sensors-22-01378-f012:**
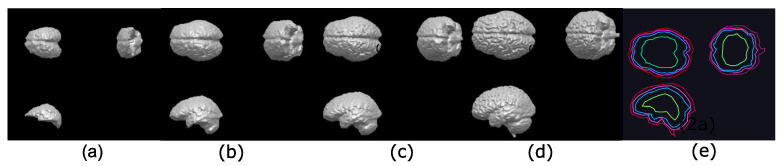
Illustration of the control in the reconstruction process by applying the hyperconnected lower leveling to the volume used in Figure 11 by varying the slope α: (**a**) α=22; (**b**) α=15; (**c**) α=10; and (**d**) α=5; (**e**) contours illustrating the reconstruction process of the brain using the leveling, with, α=22 in green, α=10 in blue, α=10 in fuchsia, and α=5 in purple.

**Figure 13 sensors-22-01378-f013:**
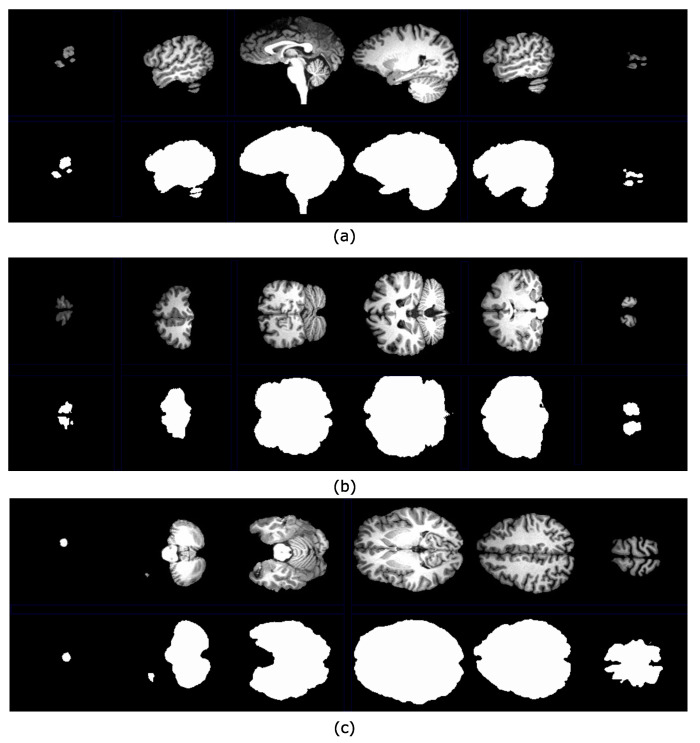
File A00028185 belongs to the NFBS repository. This volume was processed by using the algorithm presented in Figure 8. (**a**) Segmented brain slices in the sagittal plane with their corresponding ground truth images; (**b**) segmented brain slices in the coronal plane with their corresponding ground truth images; (**c**) segmented brain slices in the axial plane with their corresponding ground truth images.

**Figure 14 sensors-22-01378-f014:**
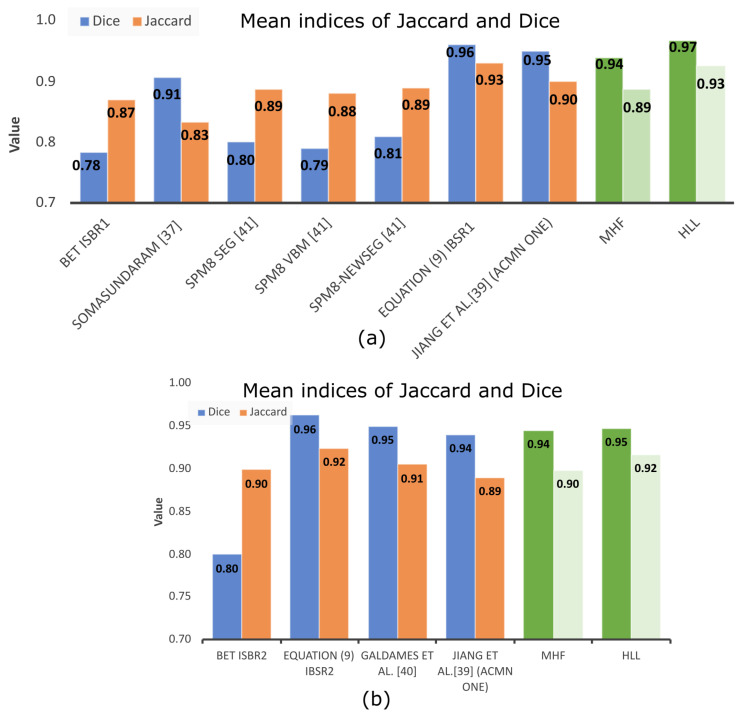
Indices comparison among the different methods reported in the current literature. (**a**) Indices of Jaccard and Dice for the methods displayed in Table 1 to the IBSR1 data set; (**b**) Indices of Jaccard and Dice for the methods displayed in Table 1 related to the IBSR2 data set.

**Figure 15 sensors-22-01378-f015:**
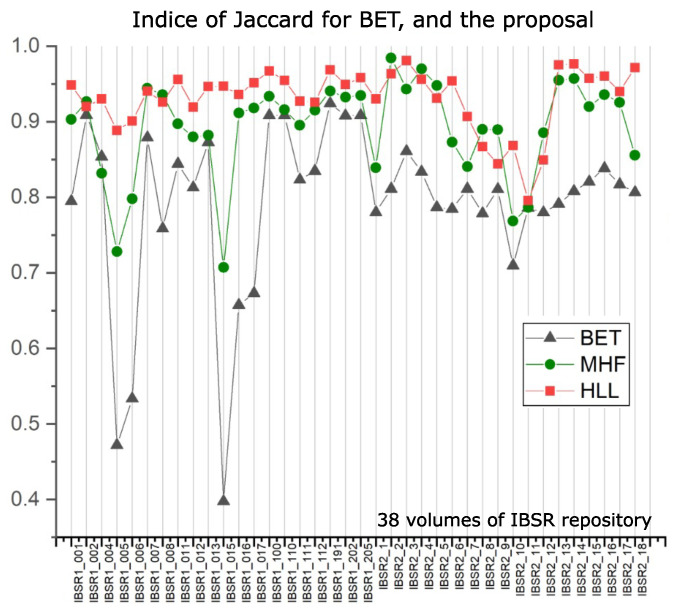
Indices of Dice and Jaccard to compare procedures BET, MHF, and HLL. Figure 8 and Figure 9 show diagrams corresponding to MHF and HLL algorithms.

**Table 1 sensors-22-01378-t001:** Jaccard and Dice indices computed for 38 volumes of an MRI obtained from the IBSR repository. The brain segmentation in this work utilizes the steps provided in Figure 8, and the computed indices are in columns MHF Jaccard and MHF Dice.

Volume	BET Jaccard	BET Dicce	MHF Jaccard	MHF Dice
IBSR1_001	0.7949	0.8857	0.9031	0.9491
IBSR1_002	0.9091	0.9524	0.9267	0.962
IBSR1_004	0.8539	0.9212	0.8318	0.9082
IBSR1_005	0.4721	0.6414	0.7281	0.8427
IBSR1_006	0.5335	0.6958	0.7981	0.8877
IBSR1_007	0.879	0.9356	0.9441	0.9713
IBSR1_008	0.7587	0.8628	0.9359	0.9669
IBSR1_011	0.8444	0.9157	0.8972	0.9458
IBSR1_012	0.813	0.8968	0.88	0.9362
IBSR1_013	0.873	0.9403	0.8822	0.9374
IBSR1_015	0.3976	0.569	0.707	0.8284
IBSR1_016	0.6575	0.7933	0.9115	0.9537
IBSR1_017	0.673	0.8045	0.9182	0.9573
IBSR1_100	0.9085	0.952	0.9337	0.9657
IBSR1_110	0.9085	0.952	0.916	0.9562
IBSR1_111	0.8233	0.9031	0.8954	0.9448
IBSR1_112	0.8347	0.9099	0.9151	0.9557
IBSR1_191	0.9243	0.9607	0.9406	0.9694
IBSR1_202	0.9082	0.9519	0.9324	0.965
IBSR1_205	0.9085	0.952	0.9347	0.9663
IBSR2_1	0.7802	0.8765	0.8392	0.9126
IBSR2_2	0.8112	0.8958	0.9843	0.9754
IBSR2_3	0.8611	0.9254	0.9432	0.9943
IBSR2_4	0.8336	0.9092	0.9698	0.9847
IBSR2_5	0.7868	0.8807	0.948	0.9733
IBSR2_6	0.7847	0.8794	0.873	0.9322
IBSR2_7	0.8113	0.8958	0.8404	0.9133
IBSR2_8	0.7787	0.8756	0.89	0.93
IBSR2_9	0.8108	0.8955	0.8895	0.9415
IBSR2_10	0.7099	0.8303	0.7685	0.8691
IBSR2_11	0.7861	0.8802	0.7865	0.8805
IBSR2_12	0.7798	0.8763	0.8855	0.9393
IBSR2_13	0.7912	0.8834	0.9548	0.9769
IBSR2_14	0.8082	0.894	0.957	0.978
IBSR2_15	0.8206	0.9014	0.92	0.9583
IBSR2_16	0.8385	0.9122	0.9359	0.9669
IBSR2_17	0.8171	0.8993	0.9256	0.9614
IBSR2_18	0.8067	0.893	0.8556	0.9222

**Table 2 sensors-22-01378-t002:** Jaccard and Dice indices computed for 38 volumes of MRI obtained from the IBSR repository. The brain segmentation in this work utilized the steps provided in Figure 9, and the computed indices are in columns HLL Jaccard and HHL Dice.

Volume	HLL Jaccard	HLL Dice	Volume	HLL Jaccard	HLL Dice
IBSR1_001	0.9255	0.9613	IBSR2_1	0.9077	0.9516
IBSR1_002	0.8981	0.9463	IBSR2_2	0.94	0.9691
IBSR1_004	0.9076	0.9515	IBSR2_3	0.9564	0.9777
IBSR1_005	0.8678	0.9292	IBSR2_4	0.9329	0.9653
IBSR1_006	0.88	0.9361	IBSR2_5	0.909	0.9523
IBSR1_007	0.9176	0.957	IBSR2_6	0.9306	0.964
IBSR1_008	0.9039	0.95	IBSR2_7	0.8855	0.9393
IBSR1_011	0.9326	0.9651	IBSR2_8	0.847	0.917
IBSR1_012	0.8976	0.946	IBSR2_9	0.825	0.9044
IBSR1_013	0.9235	0.9602	IBSR2_10	0.8483	0.9179
IBSR1_015	0.924	0.9606	IBSR2_11	0.7785	0.8754
IBSR1_016	0.9133	0.9546	IBSR2_12	0.83	0.907
IBSR1_017	0.9284	0.9629	IBSR2_13	0.9511	0.9749
IBSR1_100	0.9433	0.978	IBSR2_14	0.9523	0.9755
IBSR1_110	0.9313	0.964	IBSR2_15	0.934	0.9645
IBSR_111	0.905	0.9501	IBSR2_16	0.9368	0.9673
IBSR_112	0.9037	0.95	IBSR2_17	0.9173	0.9568
IBSR_191	0.945	0.971	IBSR2_18	0.9478	0.9732
IBSR_202	0.9262	0.962			
IBSR_205	0.935	0.9663			

**Table 3 sensors-22-01378-t003:** Jaccard and Dice indices computed for 10 volumes obtained from the neurofeedback skull-stripped (NFBS) repository.

Volume	MFL Jaccard	MFL Dice
A00028185	0.9669	0.9832
A00028352	0.9263	0.9617
A00032875	0.8360	0.9107
A00033747	0.8775	0.9348
A00034854	0.9279	0.9626
A00035072	0.9654	0.9824
A00035827	0.9653	0.9823
A00035840	0.9678	0.9836
A00037112	0.9653	0.9823
A00037511	0.8883	0.9409

**Table 4 sensors-22-01378-t004:** Jaccard and Dice indices mean values reported in the literature.

Method	Dice Average	Jaccard Average	Volumes Number
Somasundaram et al. [35]	0.9068	0.8321	20
Zhang et al. [36]	0.960	0.923	10
Jiang et al. [37](ACMN One)	0.95	0.905	38
Mendiola et al. [38] (Equation (9))	0.9645	0.9295	38
Galdames et al. [39]	0.950	0.905	18
SPM8 Seg [34]	0.8	0.888	20
SPM8 VBM [34]	0.79	0.88	20
SPM8-NewSeg [34]	0.81	0.89	20
FSL [34]	0.67	0.89	20
Brainsuite [34]	0.74	0.89	20
MHF method applied to 10 volumes of NFBS	0.96	0.92	10
**MHF method**	0.9416	0.863	38
**BET**	0.869	0.784	38
**HLL method**	0.951	0.9089	38

**Table 5 sensors-22-01378-t005:** Time in seconds to segment a volume with 5 and 60 slices using different algorithms. The experiments are conduced in a PC Intel i7, 3.4 GHz, memory ram of 16 Gb with a 64-bit operating system.

	BET	Mendiola et al. [38]	HLL Method	Jiang et al. [40]
**Average time (s)**	0.8265	450	0.9465	120

## Data Availability

Data were obtained from IBSR and NFBS repositories.

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
