# Peer review of "Hyperconnected Openings Codified in a Max Tree Structure: An Application for Skull-Stripping in Brain MRI T1"

_sensors, 2022, doi:10.3390/s22041378_

Round 1

Reviewer 1 Report

The authors presented methods for brain segmentation in MRI images. The methods are introduced with details. The results look promising. Overall quality is good for the presentation. A few suggestions which may further improve the manuscript.

  1. x and y labels and units are missing in Fig. 8.
  2. Is it intentionally to overlap (a) at the background of (b) and (f) in Fig. 8?
  3. It may be better if the authors could compare their methods with some state-of-the-art methods to prove the quality of their work.
  4. It might be better to add annotations close to the slice samples in Fig. 12-15.

Author Response

Thanks for your commentaries, below each one of them is addressed.

Reviewer#1, Concerns # 1 and #2:

  1. x and y labels and units are missing in Fig. 8.
  1. Is it intentionally to overlap (a) at the background of (b) and (f) in Fig. 8?

Author response: Taking into consideration the suggestion given by one of the reviewers, figure 8 and several equations were suppressed from this paper to clarify its content. Tables, images, and the outcome of this paper is presented in a new improved way.

Author action:  A new version of the paper is presented. Changes consisted of; 1) several equations were eliminated, 2) additional information to explain all concepts related to mathematical morphology was added, 3) data extracted from tables were graphed for clarity and comparison of Jaccard and Dice indices, 4) images and tables were selected according to their content to improve visualization of the results.

Reviewer#1, Concern # 3:

3.-It may be better if the authors could compare their methods with some state-of-the-art methods to prove the quality of their work.

Author response: Table 4 illustrates a comparison of the results extracted from our proposal with 7 different methodologies, including BET software.

Author action:  Figure 15 was added. This picture shows several graphs of Jaccard, and Dice indices taken from tables 1, 2 and 4.

Reviewer#1, Concern # 4:

4.-It might be better to add annotations close to the slice samples in Fig. 12-15.

Author response: More text was added to the article and some figure captions were further explained.

Author action:  More text was added in page 8, and some figure captions were further explained, for example Figs. 10, 11, 13.

Reviewer 2 Report

This is an interesting manuscript describing two procedures involving a maximal hyperconnected function and a hyperconnected lower leveling to segment the brain in T1 weighted MRI using new openings on a max-tree structure.

I believe this manuscript is of sufficient interest to the image processing and AI community and is appropriate for the special issue.

My concerns are minor and are listed below:

  1. The manuscript is a bit difficult to read; it may benefit from some rigorous editing.  There are a few typos as well.
  2. The figures can be of better resolutions. More detailed legends in Figures 1, 3, 4 and 11 can make it more comprehensible to the lay readers.
  3. Tables 1 and 2 are hard to follow. It may be better to condense this with Figure 9. Same goes for Table 3 and 4 and Figure 10.
  4. Table5 and 6 are too small to read.

Author Response

Thanks for your commentaries, below each one of them is addressed.

Reviewer#2, Concern # 1:  

1.-The manuscript is a bit difficult to read; it may benefit from some rigorous editing.  There are a few typos as well.  

Author response: 

A new version of the paper is presented. Changes consist of; 1) several equations were eliminated, 2) additional information to explain all concepts related to mathematical morphology was added, 3) data extracted from tables were graphed for clarity and comparison, 4) images and tables were selected according to their content to improve visualization of the results.

However, a few necessary equations were maintained to explain our proposal. The paper was revised, and concepts were explained to be easily understood by students and researchers with or without a major in mathematical morphology. Additionally, English language was revised and corrected.

Author action: We update the paper by modifying the following:

A new version of the paper is presented. Changes consist of; 1) several equations were eliminated, 2) additional information to explain all concepts related to mathematical morphology was added, 3) data extracted from tables were graphed for clarity and comparison, 4) images and tables were selected according to their content to improve visualization of the results.

Reviewer#2, Concern # 2:  

2.-The figures can be of better resolutions. More detailed legends in Figures 1, 3, 4 and 11 can make it more comprehensible to the lay readers.

Author response:   Image resolution was increased for all of them.

Author action: Image resolution was improved, and new legend was added to explain each picture.

Reviewer#2, Concerns # 3 and #4:  

3.-Tables 1 and 2 are hard to follow. It may be better to condense this with Figure 9.

4.-Same goes for Table 3 and 4 and Figure 10.

Author response:   Tables and images were condensed according to your suggestion.

Author action: Tables and images were organized according the reviewers suggestion. This situation can be observed in pages 9, 10, 11, 12 and 13 of the new version of the paper.

Reviewer#2, Concerns #5:  

5.-Table5 and 6 are too small to read.

Author response:   The size of table 5 and 6 was increased.

Author action:  The size of both tables was increased for a better visualization.

Reviewer 3 Report

The paper proposes the Skull-Stripping method applied for MRI T1 brain images. The method is based mostly on morphological operations. The results are promising, but no better than those already obtained by other researchers. However, the biggest problem of this work is the use of advanced mathematics. The concepts of "maximal hyperconnected function" and "hyperconnected lower leveling" are very complex and virtually impossible to understand for a reader without in-depth math education (the typical Sensors reader may not have such background). The paper presents as many as 25 complex formulas that have not been fully explained (which is justified by the limited volume of the work). I would suggest a significant limitation of mathematical considerations to the necessary minimum. In particular, remove information that is already known and focus on your own achievements. The merits should mainly be limited to illustrations, such as Figs. 6 and 7, to explain the operation of the proposed segmentation method. Some figures are incomprehensible, such as 4, 5 and 8 - they do not help in understanding the described method. If it not possible to remove the mathematics, please consider more suitable journal for submission.

Regarding the results obtained, please address the following issues:

Give a definition of the Dice and Jaccard coefficients.

Include in the detailed analysis the results from Jiang [39] as the dataset is as used in this study. Also, the obtained results are similar to those in [21].

Why are the values of Jaccard parameter for the proposed method much worse than those given in [21] and [39]?

Provide the operating time of the methods also for [39].

Author Response

Thanks for your commentaries, below each one of them is addressed.

Reviewer#3, Concern # 1:

  1. - Give a definition of the Dice and Jaccard coefficients.

Author response:  The mathematical definition of Jaccard and Dice indices was included. On the same page, a brief explanation of their advantages and disadvantages was added.

Author action:  The mathematical definition of Jaccard and Dice indices with a brief explanation about their use was added in page 8.

________________________________________

Reviewer#3, Concern # 2:

Include in the detailed analysis the results from Jiang [39] as the dataset is as used in this study. Also, the obtained results are similar to those in [21].

Author response:  The proposal given in Jian [39] is now explained in the new version of the paper. 

Author action:  The following text was added in pages 8 and 13:

In Jiang et al. [36], authors use a nonlinear speed function in the hybrid level set model to eliminate boundary leakage. When using the method, an active contour neighborhood model is applied iteratively slice by slice until the neighborhood of the brain boundary is obtained. The results show high values for the computed indices because of the edges enhancement. However, there are two problems; 1) the brain extraction requires a semi-global understanding of the image, and 2) the weak boundaries between the brain tissues and surrounding tissues.

________________________________________

Reviewer #3 , Concern # 3:

Why are the values of Jaccard parameter for the proposed method much worse than those given in [21] and [39]?

Author response:  From table 4 and Fig. 14, results indicate minimal differences among the methods  [35][36][37][38] and the proposal provided in the paper.  The disadvantage of the proposal introduced here is that before loading the flat areas to the maxtree, a smoothing filtering is performed to reduce the number of regions that are loaded to the maxtree, producing the fusion of elongated and narrow regions. Furthermore, comparing separately the indexes of the averages of the 18 (IBSR1) and 20 (IBSR2) volumes, the method proposed here is superior to that of Jian in the IBSR1 repository.  

Author action:  Next text was added in page 13.

From table 4 and Fig. 14, results indicate minimal differences among the methods [35][36][37][38] and the proposal provided in the paper. The disadvantage of the proposal introduced here is that before loading the flat areas to the maxtree, a smoothing filtering is performed to reduce the number of regions that are loaded to the maxtree, producing the fusion of elongated and narrow regions. Furthermore, comparing separately the indexes of the averages of the 18 (IBSR1) and 20 (IBSR2) volumes, the method proposed here is superior to that of Jian in the IBSR1 repository.  

________________________________________

Reviewer #3, Concern # 4:

Provide the operating time of the methods also for [39].           

Author response:  The execution time of [39 ]  can be found in table [5];

Author action:  Next text was added in page 13.

The execution time  of [39 ]   can be found in table [5].

Round 2

Reviewer 3 Report

First part of my review was not addressed at all.  

Any introduces changes and amendments should br clearly indicated in the revised version of the manuscript.

Author Response

Dear reviewer, we apologize for not having sent the complete answer to your comments; however, we send them and are attentive to your observations.

Reviewer#3 Comment

The paper proposes the Skull-Stripping method applied for MRI T1 brain images. The method is based mostly on morphological operations. The results are promising, but no better than those already obtained by other researchers. However, the biggest problem of this work is the use of advanced mathematics. The concepts of "maximal hyperconnected function" and "hyperconnected lower leveling" are very complex and virtually impossible to understand for a reader without in-depth math education (the typical Sensors reader may not have such background). The paper presents as many as 25 complex formulas that have not been fully explained (which is justified by the limited volume of the work). I would suggest a significant limitation of mathematical considerations to the necessary minimum. In particular, remove information that is already known and focus on your own achievements. The merits should mainly be limited to illustrations, such as Figs. 6 and 7, to explain the operation of the proposed segmentation method. Some figures are incomprehensible, such as 4, 5 and 8 - they do not help in understanding the described method. If it not possible to remove the mathematics, please consider more suitable journal for submission.

Author response: Thanks for your commentaries. The response to your comments is given below:

Reviewer#3, Comment 1.-The paper proposes the Skull-Stripping method applied to MRI T1 brain images. The method is mostly based on morphological operations. The results are promising, but no better than those already obtained by other researchers.

Author response: We made an extension of the results presented in [38], where the algorithm's execution time is higher to segment MRI volumes.  It is noteworthy that our proposal overcomes the execution time found in [37][40],  but with similar Jaccard and Dice indices, since there is no significative difference between such indices.

  1. Jiang, S.; Zhang, W.; Wang, Y.; Chen, Z. Brain extraction from cerebral MRI volume using a hybrid level set based active contour neighborhood model. Biomedical engineering online 2013, 12, 1–18.

  1. Mendiola-Santibañez, J.D.; Gallegos-Duarte, M.; Arias-Estrada, M.O.; Santillán-Méndez, I.M.; Rodríguez-Reséndiz, J.; Terol-Villalobos, I.R. Sequential application of viscous opening and lower leveling for three-dimensional brain extraction on magnetic resonance imaging T1. Journal of Electronic Imaging 2014, 23, 1 – 14. doi:10.1117/1.JEI.23.3.033010.
  2. Jiang, S.; Wang, Y.; Zhou, X.; Chen, Z.; Yang, S. Brain Extraction Using Active Contour Neighborhood-Based Graph Cuts Model. Symmetry 2020, 12, 559.

%%%%%%%%%%%%%%%%%%%%%%%%%%%%%%%%%%%%%%%%%%%%%%%%%%%%%%%%%

Reviewer#3, Comment 2.  -The concepts of "maximal hyperconnected function" and "hyperconnected lower leveling" are very complex and virtually impossible to understand for a reader without in-depth math education (the typical Sensors reader may not have such background).

Author response:  Hyperconnectivity means connected components. To better explain the concept, the following text was added to the paper in section 2.5 on page 4:

Serra introduced the hyperconnectivity concept [23], which permits working with  joined or overlapped components. Figure 4 helps to understand this notion for the 2D case.

Notice that the eyes link the brain and the skull in Fig. 4(a), i.e., they are hyperconnected  because they form a unique component. Figure 4(b) presents the regional maxima obtained  from Fig. 4(a). Each of these maxima is located on the brain, eyes, white matter, or skull.

%%%%%%%%%%%%%%%%%%%%%%%%%%%%%%%%%%%%%%%%%%%%%%%%%%%%%%%%%%%%%%%%%%%%%%%%%%%%%%%%%%%%%%%%%%%%%%%%%%%%%%%%%%%%%%

 Reviewer#3, Comment 3The paper presents as many as 25 complex formulas that have not been fully explained (which is justified by the limited volume of the work). I would suggest a significant limitation of mathematical considerations to the necessary minimum. In particular, remove information that is already known and focus on your own achievements.

Author response:  The new version of the paper presents fewer equations (11 equations) , and we reduce information considerably.

%%%%%%%%%%%%%%%%%%%%%%%%%%%%%%%%%%%%%%%%%%%%%%%%%%%%%%%%%%%%%%%%%%%%%%%%%%%%%%%%%%%%%%%%%%%%%%%%%%%%%%%%%%%%%%

Reviewer#3, Comment 4.  The merits should mainly be limited to illustrations, such as Figs. 6 and 7, to explain the operation of the proposed segmentation method. Some figures are incomprehensible, such as 4, 5 and 8 - they do not help in understanding the described method. If it not possible to remove the mathematics, please consider more suitable journal for submission.

Author response:  A new version of the paper is presented. Changes consist of; 1) several equations were eliminated, 2) additional information to explain all concepts related to mathematical morphology was added, 3) data extracted from tables were graphed for clarity and comparison, 4) images and tables were selected according to their content to improve visualization of the results.

To make the paper easier to understand, images and tables were organized according to their contents. Figures 4 and 8 were suppressed to avoid confusion, while Fig 5 maintained the same position number. Fig 5 was explained on pages 5 and 6  of the new version of the paper. The following text was added:

Figure 5 exemplifies this situation. Figure 5 (a) shows two overlapping functions g1 and  g2. For the MRI case, g1 represents a maximum on the brain, and g2 is a maximum on the  skull; however, regions under the intersection of both functions indicate that the brain  and the skull overlap, i.e., they are connected or hyperconnected. Fig. 5(b) illustrates the supremum between g1 and g2. Note that it is impossible to recover g1 or g2 from g1. This is what we visualize in reality; our eyes would observe how the brain and the skull  appear in two places in a certain slice; nevertheless, lower slices connect them.  The reconstructed functions correspond to gM1 and gM2, displayed in Figs. 5(c) and 5(d).  These images come from an individual reconstruction using each maximum computed  from Fig. 5(b).  Figs. 5(c) and 5(d) illustrate how to detect the markers to separate brain and  skull, and Equation 6 represents it formally. The maximum to be treated is selected and subsequently reconstructed using the transformation by reconstruction R. In Fig. 5(e), functions g1 and g2 do not overlap; hence both functions can be retrieved. However, this is not what really happens in image segmentation.

%%%%%%%%%%%%%%%%%%%%%%%%%%%%%%%%%%%%%%%%%%%%%%%%%%%%%%%%%%%%%%%%%%%%%%%%%%%%%%%

Reviewer#3, Concern # 5:

  1. - Give a definition of the Dice and Jaccard coefficients.

Author response:  The mathematical definition of Jaccard and Dice indices was included. On the same page, a brief explanation of their advantages and disadvantages was added.

Author action:  The mathematical definition of Jaccard and Dice indices with a brief explanation about their use was added in page 8.

%%%%%%%%%%%%%%%%%%%%%%%%%%%%%%%%%%%%%%%%%%%%%%%%%%%%%%%%%%%%%%%%%%%%%%%%%%%%%%%

Reviewer#3, Concern # 6:

Include in the detailed analysis the results from Jiang [39] as the dataset is as used in this study. Also, the obtained results are similar to those in [21].

Author response:  The proposal given in Jian [39] is now explained in the new version of the paper. 

Author action:  The following text was added in pages 8 and 13:

In Jiang et al. [36], authors use a nonlinear speed function in the hybrid level set model to eliminate boundary leakage. When using the method, an active contour neighborhood model is applied iteratively slice by slice until the neighborhood of the brain boundary is obtained. The results show high values for the computed indices because of the edges enhancement. However, there are two problems; 1) the brain extraction requires a semi-global understanding of the image, and 2) the weak boundaries between the brain tissues and surrounding tissues.

%%%%%%%%%%%%%%%%%%%%%%%%%%%%%%%%%%%%%%%%%%%%%%%%%%%%%%%%%%%%%%%%%%%%%%%%%%%%%%%

Reviewer #3 , Concern # 7:

Why are the values of Jaccard parameter for the proposed method much worse than those given in [21] and [39]?

Author response:  From table 4 and Fig. 14, results indicate minimal differences among the methods  [35][36][37][38] and the proposal provided in the paper.  The disadvantage of the proposal introduced here is that before loading the flat areas to the maxtree, a smoothing filtering is performed to reduce the number of regions that are loaded to the maxtree, producing the fusion of elongated and narrow regions. Furthermore, comparing separately the indexes of the averages of the 18 (IBSR1) and 20 (IBSR2) volumes, the method proposed here is superior to that of Jian in the IBSR1 repository.  

Author action:  Next text was added in page 13.

From table 4 and Fig. 14, results indicate minimal differences among the methods [35][36][37][38] and the proposal provided in the paper. The disadvantage of the proposal introduced here is that before loading the flat areas to the maxtree, a smoothing filtering is performed to reduce the number of regions that are loaded to the maxtree, producing the fusion of elongated and narrow regions. Furthermore, comparing separately the indexes of the averages of the 18 (IBSR1) and 20 (IBSR2) volumes, the method proposed here is superior to that of Jian in the IBSR1 repository.  

%%%%%%%%%%%%%%%%%%%%%%%%%%%%%%%%%%%%%%%%%%%%%%%%%%%%%%%%%%%%%%%%%%%%%%%%%%%%%%%

Reviewer #3, Concern # 7:

Provide the operating time of the methods also for [39].       

Author response:  The execution time of [39 ]  can be found in table [5];

Author action:  Next text was added in page 13.

The execution time  of [39 ]   can be found in table [5].
